# Effect of Soil Type: Qualitative and Quantitative Analysis of Phytochemicals in Some Browse Species Leaves Found in Savannah Biome of South Africa

**DOI:** 10.3390/molecules27051462

**Published:** 2022-02-22

**Authors:** Humbelani Silas Mudau, Hilda Kwena Mokoboki, Khuliso Emmanuel Ravhuhali, Zimbili Mkhize

**Affiliations:** 1Department of Animal Science, School of Agricultural Sciences, Faculty of Natural and Agricultural Sciences, North-West University, Mmabatho 2735, South Africa; hilda.mokoboki@nwu.ac.za; 2Food Security and Safety Niche Area, Faculty of Natural and Agricultural Sciences, North-West University, Mmabatho 2735, South Africa; 3Department of Chemistry, School of Chemical and Physical Sciences, Faculty of Natural and Agricultural Sciences, North-West University, Mmabatho 2735, South Africa; zimbili.mkhize@nwu.ac.za

**Keywords:** phytochemical, browse species, quantitative screening, qualitative screening, livestock, bioactive compounds

## Abstract

In semi-arid regions, browse plant species are used as feed and for medicinal purposes for both animals and humans. The limitation of the utilization of these species to medicinal purposes or as feed for livestock is a lack of knowledge on the concentration level of phytochemicals and other bioactive compounds found in these plants. The study sought to assay the qualitative and quantitative bioactive constituents of some browse species found in the savannah biome of South Africa, viz. *Adansonia digitate*, *Androstachys johnsonii*, *Balanites maughamii*, *Berchemia discolor*, *Berchemia zeyheri*, *Bridelia mollis hutch*, *Carissa edulis*, *Catha edulis*, *Colophospermum mopane*, *Combretum Imberbe*, *Combretum molle*, *Combretum collinum*, *Dalbergia melanoxylon*, *Dichrostachys cinerea*, *Diospros lycioides*, *Diospyros mespiliformis*, *Euclea divinorum*, *Flueggea virosa*, *Grewia flava*, *Grewia flavescens*, *Grewia monticola*, *Grewia occidentalis*, *Melia azedarach*, *Peltophorum africanum*, *Prosopis velutina*, *Pseudolachnostylis maprouneifolia*, *Pterocarpus rotundifolius*, *Schinus molle*, *Schotia brachypetala*, *Sclerocarya birrea*, *Searsia lancea*, *Searsia leptodictya*, *Searsia pyroides*, *Senegalia caffra*, *Senegalia galpinii*, *Senegalia mellifera*, *Senegalia nigrescens*, *Senegalia polyacantha*, *Strychnos madagascariensis*, *Terminalia sericea*, *Trichilia emetic*, *Vachellia erioloba*, *Vachellia hebeclada*, *Vachellia karroo*, *Vachellia nilotica*, *Vachellia nilotica* subsp. *Kraussiana*, *Vachellia rechmanniana*, *Vachellia robusta*, *Vachellia tortilis*, *Vachellia tortilis* subsp. *raddiana*, *Vangueria infausta*, and *Ziziphus mucronata*. These browse species’ leaf samples were harvested from two provinces (Limpopo and North-West) of South Africa. The Limpopo province soil type was Glenrosa, Mispah, and Lithosols (GM-L), and the soil types in the North-West Province were Aeolian Kalahari Sand, Clovelly, and Hutton (AKS-CH). The harvested browse samples were air dried at room temperature for about seven days and ground for analysis. The methanol and distilled water extracts of the browse species leaves showed the presence of common phytoconstituents, including saponins, flavonoids, tannins, phenols, cardio glycosides, terpenoids, and phlobatannins, as major active compounds in browse species leaves. In the quantitative analysis, phytochemical compounds, such as soluble phenols, insoluble tannins, and condensed tannins, were quantified for common species found in both sites. Two-way ANOVA and multivariate analysis were used to test soil type and species effect on soluble phenols, insoluble tannins, and condensed tannins of woody species. *Dichrostachys cinerea* (0.1011% DM) in GM-L soil type and *Z. mucronata* (0.1009% DM) in AKS-CH soil type showed the highest (*p* < 0.05) concentration of soluble phenols. In AKS-CH soil type, *D. cinerea* (0.0453% DM) had the highest insoluble tannins concentration, while *V. hebeclada* had the lowest (0.0064% DM) insoluble tannins content. *Vacchelia hebeclada* had lower (*p* < 0.05) condensed tannins concentration levels than all other browse plants in both soil types. Under multivariate analysis tests, there was a significant effect (*p* < 0.001) of soil type, species, and soil type x species interaction on soluble phenols, insoluble tannins, and condensed tannins of woody species. In this study, most of the woody species found in GM-L soil type showed a lower amount of tannins than those harvested in AKS-CH soil type. There is a need to identify the amount of unquantified phytochemicals contained in these browse species and valorize the high-bioactive-compound browse species to enhance and maximize browsing of these browse species for animal production.

## 1. Introduction

Savannah woody species play an important role as the cheapest protein source for livestock in semi-arid regions [1]. Some of these plants are deciduous and semi-succulent. They can retain green leaves in certain seasons of the year, making them advantageous in responding to feed shortages during dry seasons, when the herbaceous species and other annual grasses wilt [2,3]. These plants’ contributions to multi-purpose tasks (protein and medicinal sources) have been documented [4,5,6].

The chemical composition of browse species varies with plant species, growth, plant parts, soil type, temperature, altitude, light, and other anthropogenic factors, such as herbivory [7,8,9,10]. Some of these woody species contain plant secondary metabolites (phytochemicals) which are anti-nutritional elements, which may negatively affect an animal’s health and nutrition [11,12]. A large number of bioactive compounds, including phenols, flavonoids, tannins, coumarin, lignin, terpenes, saponins, phlobatannins, and glycosides, are usually scattered in plant species [13]. Mazid et al. [14] pointed out that phenolic compounds are known to be abundant plant secondary metabolites in browse species which can contribute up to half of their organic matter.

Furthermore, Liu et al. [15] highlighted that phenolic compounds, structurally, are made up of aromatic benzene rings with more than one hydroxyl group, arranged from lower to higher polymerization levels. According to Mueller-Harvey [16], tannins, flavonoids, and phenolic acids are phenolic compounds which are abundantly constituted in savannah woody species, and they play a significant role in animal nutrition, health, and performance [17]. Flavonoids are a group of secondary metabolites found in plants with a lower molecular weight, and they are constituted with major classes, such as flavonols, flavones, and isoflavones.

Plant secondary metabolites (phytochemicals) offer important pharmacological and biological properties, including hypoglycemic, anti-inflammatory, antioxidant, anti-bacterial, and anti-carcinogenic properties, which can fight against oxidative cell damage [18,19,20]. These bioactive compounds are also essential sources for proper plant growth and defensive mechanisms against infection and plant cell damage. Salem et al. [21] indicated that the isolation and identification of browse plant phytochemicals have long been a difficult challenge for researchers. Different plant species can have variation in secondary metabolites produced through biosynthesis [22]. Though there are numerous research studies that have been conducted worldwide with special focus on phytochemical medicinal plants [23,24,25,26,27], little scientific validation has been carried out for browse species phytochemicals found in most semi-arid areas of South Africa. There is a need to perform screening of phytochemicals in order to establish the concentration levels of these browse species. This is a valuable step to serve as a guiding tool to identify and manipulate those browse species that might be toxic to ruminants due to high concentration of certain bioactive compounds. Therefore, the current study was designed to evaluate the existence of phytochemical components and assess the qualitative and quantitative aspects of phytochemicals present in browse species extracts. The study hypothesized that there was variation in the qualitative and quantitative aspects of phytochemicals present in browse plant species, given the therapeutic value of these widely accessible browse plant species. Hopefully, the study will contribute to the improvement of livestock production in savannah areas of South Africa.

## 2. Results

### 2.1. Preliminary Qualitative Phytochemical Analysis

Results of phytochemical screenings of browse species found in GM-L soil type are shown in Table 1. Phytochemical compounds such as saponins, flavonoids, tannin, phenols, glycosides, terpenoids, and phlobatannins were screened and observed in methanol and distilled water extracts. Phlobatannins were found to be absent in distilled water extracts for all browse species harvested in both soil types. Terpenoids were present in both methanol and distilled water extracts, except for *S. galpinii*, which was found to be negative in the distilled water extract. Negative results for tannins and phenols were found in most browse leaves from distilled water extracts. *Vachellia hebeclada* showed negative results for phlobatannins, glycosides, phenols, and tannins in both methanol and distilled water extracts. Saponins and flavonoids were present in all browse species from distilled water extracts, except for *C. edulis*, *T. sericea*, and *V. tortilis* subsp. *raddiana*.

In Table 2, for AKS-CH soil type, terpenoids were present in both methanol and distilled water extracts, except only for *V. nilotica* subsp. *Kraussiana*, which were absent in the water extract. Saponins were not present only in *S. molle* when using both methanol and distilled water extracts. *Senegalia galpinii* and *V. hebeclada* were found negative in tannins, phenols, glycoside, and phlobatannins in both methanol and distilled water extracts. Flavonoid, saponins, tannins, and phenols were present in most browse species from methanol extracts.

### 2.2. Quantitative Phytochemical Analysis

Table 3 showed the findings of the effect of species and soil type on the soluble phenols, insoluble tannins, and condensed tannins of browse plants leaves. There were significant differences (*p* < 0.05) observed on the effect of species, soil type, and the interaction between species and soil type on soluble phenol, insoluble tannin, and soluble condensed tannin concentrations of browse leaves found in communal rangelands. In GM-L soil type, *V. nilotica* subsp. *Kraussiana* and *T. sericea* had a higher concentration of insoluble tannins (0.0386% DM), while in AKS-CH soils, *D. cinerea* had a higher concentration of insoluble tannins (0.0453% DM) when compared to all other species in the same harvesting site. *Melia azedarach*, *D. cinerea*, *S. mole*, *S. leptodictya*, *V. hebeclada*, and *V. karroo* in GM-L soil type had higher insoluble tannin concentration when compared to all other species in the same harvesting site. In both sites, *V. hebeclada* browse leaves had the lowest (*p* < 0.05) insoluble tannin (0.0041% DM, GM-L and 0.0064% DM, AKS-CH soil type) and condensed tannin (0.0137 AU_550_/200 mg, GM-L and 0.0138 AU_550_/200 mg, AKS-CH soil type) concentrations. In both sites, *V. hebeclada* browse leaves had the lowest (*p* < 0.05) concentration levels of soluble phenols (Limpopo—0.0160% DM) and (North-West—0.0334% DM). Within GM-L and AKS-CH soil types, *V. tortilis* has shown no significant difference in soluble phenolic content. Again, in the concentration of CTs, *Z. mucronate*, *V. hebeclada*, *P. africanum*, and *T. sericea* did not show any significant differences.

The results on multivariate tests on the effect of species and soil type on the soluble phenols, insoluble tannins, and condensed tannins of browse plants leaves are shown on Table 4. All of the multivariate tests displayed significant differences (*p* < 0.001) on the effect of species, soil type, and the interaction between species and soil type on soluble phenol, insoluble tannin, and condensed tannin concentration of browse leaves found in communal rangelands.

## 3. Discussion

### 3.1. Phytochemical Screening of Browse Species

In almost all cases, the methanol extract showed the presence of a rich variety of secondary metabolites when compared to the distilled water extract. This may be due to the fact that methanol extract tends to dissolve many organic compounds more quickly than in water extract. Again, Paulsamy and Jeeshna [28] highlighted that methanolic solvent tends to draw a higher variety of plant compounds than other solvents. Ghasemzadeh et al. [29] explained that the variation in browse species’ nature and polarity level plays a major role in extracting plant secondary metabolites. Similar results were also reported by Shen and Shao [30] and Ingle and Padole [27].

Plant secondary metabolites, which are chemical compounds responsible for biological activities, are commonly found in browse plants [31]. From phytochemical screening in the current study, we observed that the methanol and distilled water extracts from browse species gave positive results in the major phytoconstituent compounds. In this study, *Catha edulis* leaves from GM-L and *S. molle* leaves from AKS-CH soil type reported negative in saponin concentration using both methanol and distilled water extracts. The frothing tests were used in the present study to confirm the presence of saponins in the methanol and distilled water extracts on browse species leaves. Okwu [32] indicated that saponins have the distinctive properties of precipitating and coagulating red blood cells and also help in regulating the immune response. These results are in line with those reported by Chaudhary et al. [33] that saponins induce the degradation of cell wall enzymes and the leakage of proteins from the cell.

In this study, flavonoids were found to be moderately present when tested with distilled water extract as compared to those tested with methanol extracts, which were more abundant, and these results are in agreement with those reported by Ajiboye et al. [34] and Senguttuvan et al. [24]. Flavonoids are polyphenolic compounds that are water-soluble and are extensively dispersed in browse plants species, with over 5000 flavonoids discovered [35]. Flavonoids have anti-allergic, anti-inflammatory, tumor inhibitory, antiviral, and anti-cancer properties in addition to antioxidant activities [36,37]. These effects have been associated with health-promoting properties and a wide range of chemical and biological functions, including vasoprotection and free radical scavenging, which can prevent oxidative cell damage [36]. Rauf et al. [38] also reported that flavonoid-rich browse plants are diuretics, whereas others are anti-spasmodic and antibacterial.

Browse plants usually generate phenolics, which are aromatic benzene ring molecules containing one or more hydroxyl groups, mostly to protect plants against stress [39]. Tannin accumulation levels on browse plant species play a critical role in animal health, because most plants frequently produce these secondary metabolites either in high or low volume. Hoste et al. [40] indicated that tannins as polyphenolics in natural plants may have detrimental effects on animals consuming high amounts (>6%) of tannins in their dietary feed. In this study, tannins and phenols were found negative in *V. hebeclada* from both solvent (methanol and distilled water) extracts. *Vachellia hebeclada* can be used as a beneficial browse plant for animal consumption due to the fact that it has low levels of tannins and phenols, which are known to be anti-nutritional to animal diet when above 6 g/kg of the total intake diet [41]. From the present study, negative results for tannins and phenols were found in most browse leaves from the distilled water extract. Nevertheless, tannins and phenols are not only considered to have detrimental effects, but they also have some beneficial effects that have been documented. Okwa and Josiah [42] reported that tannins have a critical role in promoting healing of wounds and inflamed mucous membranes.

In this study, cardiac glycosides were present, except in *V. hebeclada* (GM-L and AKS-CH soil types) and *M. azederach* (GM-L soil type) in both methanol and distilled water extracts. These results may be influenced by browse plant genetic makeup, because polarity did not show any influence in the results reported by Senguttuvan et al. [24] when using methanol and water solvents to test cardiac glycosides. Konstantinou et al. [43] indicated that cardiac glycosides have a direct effect on the heart; they normally aid in restoring a failing heart by maintaining the contraction rate and heart strength. Browse plants rich in glycosides are known to strengthen the immune system, making them useful as dietary supplements. In general, glycosides are present in every woody plant species and play a significant role in therapeutic efficacy. From the present study, terpenoids were found to be present in both methanol and distilled water extracts, except for *S. galpinii* (GM-L soil type) and *V. nilotica* subsp. *krassianna* (AKS-CH soil type), which were found to be negative in the distilled water extract. Several authors, such as Wagner and Elmadfa [44] along with Rabi and Bishayee [45], have found that terpenoids have great potential as beneficial remedies for treating and preventing various ailments in both livestock and humans. Again, Wagner and Elmadfa [44] also highlighted that terpenoids have several functions, such as anti-parasitic, antifungal, antimicrobial, and anti-inflammatory properties. Furthermore, terpenoids tend to have insecticidal characteristics, and they can be used as protective chemicals in the storage of agricultural products.

Phlobatannins, in the current study, were found to be absent when using distilled water extract for all woody species leaves in both soil types. These findings may be influenced by the polarity level of water, which was not suitable for drawing phlobatannins in the selected browse leaves. Setchell [46] indicated that phlobatannins in browse plant species have astringent properties, such as anti-inflammation and antiviral as well as antioxidant activities. Moreover, the antimicrobial, antifungal, and anti-parasitic activities of woody species as well as the presence or dispersion of phytochemical compounds appeared to be partly dependent on the host plant species.

### 3.2. Phytochemical Quantification of Browse Species

Secondary plant compounds, which included tannins and phenolics, normally serve important roles in tropical animal productivity, because animals in semi-arid regions rely on woody species as a source of daily feed, and this browse forage tends to accumulate high-concentration levels of phytochemical compounds [41]. In this study, the findings of phytochemical compounds (soluble phenols, insoluble tannins, and condensed tannins) were significantly different among woody species harvested in both sites. Ravhuhali et al. [47] highlighted that woody species found in various locations had a wide range of chemical constituents. Browse species, soil type, and their interaction had an impact on the concentration levels of soluble phenolic (SPhs), insoluble tannins (ITs), and condensed tannins (CTs). The concentration of SPhs on woody species leaves from both soil types was between 0.0160 and 0.1011% DM, and those woody species in AKS-CH soil type had higher values than those from GM-L soil type. This might be due to different altitudes and soil profiles, because plants growing in the poor soil profiles are known to accumulate high levels of phenolics [48], and, in this study, AKS-CH soil type is assumed to have a poor soil profile when compared to GM-L soil type. Reed [49] highlighted that soluble phenols have a moderate impact on ruminant nutrition when compared to tannins. The leaves of *V. hebeclada* had the least soluble phenols in both sites. These results might have been influenced by plant genetic make-up. According to Ibrahim et al. [50], plant genetic make-up has a large impact on concentration levels of phenolics, way more than other abiotic elements (altitude, temperature, light, and soil type) that aid in the secretion of bioactive compounds in woody plants. *Schinus molle* leaves from AKS-CH soil type had the highest soluble phenolics (0.1000% DM), while in GM-L soil type they had the least soluble phenolics (0.0377% DM). These results could be due to herbivorous activities, as browse plants trees tend to produce and secrete their defensive chemicals throughout the plants’ parts as a form of defense mechanism against herbivorous activities [51]. This might warrant the investigation to assess the most preferred plant by livestock ruminants in the study areas. Mnisi and Mlambo [3] highlighted that dietary intake and digestibility of plant-based feed for livestock can be determined by the number of phenolic compounds constituted in the feed diet.

The concentration of insoluble tannins (ITs) in this study ranged from 0.0037% DM (*S. leptodictya*) to 0.0386% DM (*T. sericea* and *V. nilotica* subsp. *krassianna*) in GM-L soil type, and in AKS-CH soil type the levels ranged from 0.0064% DM (*V. hebeclada*) to 0.0453% DM (*D. cinerea*). Within GM-L soil type, *S. leptodictya*, *V. hebeclada*, *S. molle*, and *M. azedarach* had the lowest insoluble tannins content. With these results, it can be suggested that these browse species can be regarded as a suitable feed source for ruminants due to their lower tannin concentration [52,53,54].

In this study, condensed tannin (CT) concentration was influenced by species and their environments. *Vachellia hebeclada*, *M. azederaach*, *S. molle*, and *S. leptodictya* concentration levels of condensed tannins in this study are similar to the findings reported by Mlambo et al. [55] (3–6%) on the same woody species and this range is regarded as the best concentration level for herbivores. The highest concentration of CTs reported by Mokoboki et al. [11] was in *V. karroo* leaves, which agrees with the findings of this study. The majority of the selected woody species, regardless of their harvesting sites, had a lower concentration of CTs in this study, which is likely considered to be nutritionally safe. Makkar and Singh [56] indicate that the presence of condensed tannins in large quantities (>4–9%) in fiber fractions may influence their digestion kinetic characteristics, causing the link between kinetic measures and fiber composition to be muddled. This is in line with the findings reported by Foo et al. [57], Dube et al. [58], and Mlambo et al. [41], who highlighted that the concentration of CTs accumulated in the animal diet tends to interfere with intake and digestion, while very low concentration has a beneficial effect in ruminants. The variance in condensed tannins content across plant species might be related to genetic make-up, altitude, and herbivory, which results in different response mechanisms to spatial variation [47,59,60]. However, to improve the utilization of tanniniferous browse species, tannin-inactivating treatments are required. Various studies have already indicated the use of PEG (polyethylene glycol) and WA (wood ash) to deactivate the protein tannin bonds on various plant parts as a remedy to high-tannin browse plants [61,62,63].

The results showed that with the use of multivariate analysis tests, a significant effect of soil type, species, and interaction between soil types and browse species on soluble phenols, insoluble tannins, and condensed tannins of woody species. In this study, CTs, ITs, and SPhs concentration was influenced by species and their environments. Indeed, species from different locations tend to have variation in concentration level, and related results were also reported by Schleppi et al. [64] and Suárez et al. [65]. The positive view on the interaction between biotic and abiotic components with living organisms was also highlighted by Hervé et al. [66].

## 4. Materials and Methods

### 4.1. Description of the Harvesting Sites

#### 4.1.1. Limpopo

The harvesting sites were in the Thulamela and Makhado local municipalities, with a probably high number of livestock, such as goats, sheep, donkeys, and cattle that completely rely on these communal rangelands. The browse tree species were harvested in Lamvi (22°40′21″ S, 30°45′26″ E and 639 m above sea level) and Mutele (22°28′35″ S, 30°50′24″ E and 339 m above sea level) in Thulamela municipality and Mpheni (23°08′10″ S, 30°03′18″ E and 808 m above sea level) village communal area of Makhado municipality in Limpopo. The communal areas have Glenrose, Mispah, and Lithosols (GM-L) soil type, with reddish or brown sandy to loamy soil structure. The vegetation type in the selected communal areas is Soutpansberg Mountain Bushveld and Makuleke Sandy Bushveld vegetations [67]. The study site temperatures range from 13 °C to 34 °C and with rainfall range between 200 mm to 500 mm annually, respectively.

#### 4.1.2. North-West Sites

The North-West province harvesting sites were Mahikeng and Ratlou local municipalities, with high numbers of goats, sheep, and cattle grazing in these communal areas. In Mafikeng, the browse tree leaves were harvested from Six Hundred (25°42′43″ S, 25°37′32″ E and 1300 m above sea level) and Tsetse (25°44′07″ S, 25°39′40″ E and 1296 m above sea level), with both constituted with clay-loamy soil structure, and, meanwhile, Loporung (25°45′37″ S, 24°59′54″ E and 1162 m above sea level), with red-brown sandy soil structure, is situated in Ratlou municipality in North-West Province, South Africa. These communal areas have Aeolian Kalahari Sand, Clovelly, and Hutton (AKS-CH) soil type, based in Mafikeng Bushveld, Eastern Kalahari Bushveld, and Thornveld vegetation types [67]. These areas receive mean annual rainfall varying from a minimum of 400 mm to 500 mm. The study sites have winter seasons with sunny dry days and cold nights with an annual average temperature ranging between 2 °C and 36 °C, with a rainfall ranging from 250 mm to 450 mm annually, respectively.

### 4.2. Sample Identification and Collection of Browse Plants Leaves

Randomly fresh browse leaves from five trees per plant species were selected, such as *Adansonia digitate*, *Androstachys johnsonii*, *Balanites maughamii*, *Berchemia discolor*, *Berchemia zeyheri*, *Bridelia mollis hutch*, *Carissa edulis*, *Catha edulis*, *Colophospermum mopane*, *Combretum Imberbe*, *Combretum molle*, *Combretum collinum*, *Dalbergia melanoxylon*, *Dichrostachys cinerea*, *Diospros lycioides*, *Diospyros mespiliformis*, *Euclea divinorum*, *Flueggea virosa*, *Grewia flava*, *Grewia flavescens*, *Grewia monticola*, *Grewia occidentalis*, *Melia azedarach*, *Peltophorum africanum*, *Prosopis velutina*, *Pseudolachnostylis maprouneifolia*, *Pterocarpus rotundifolius*, *Schinus molle*, *Schotia brachypetala*, *Sclerocarya birrea*, *Searsia lancea*, *Searsia leptodictya*, *Searsia pyroides*, *Senegalia caffra*, *Senegalia galpinii*, *Senegalia mellifera*, *Senegalia nigrescens*, *Senegalia polyacantha*, *Strychnos madagascariensis*, *Terminalia sericea*, *Trichilia emetic*, *Vachellia erioloba*, *Vachellia hebeclada*, *Vachellia karroo*, *Vachellia nilotica*, *Vachellia nilotica* subsp. *Kraussiana*, *Vachellia rechmanniana*, *Vachellia robusta*, *Vachellia tortilis*, *Vachellia tortilis* subsp. *raddiana*, *Vangueria infausta*, and *Ziziphus mucronata*. These browse species were harvested from two different (Limpopo and North-West) provinces of South Africa, semi-arid areas under savannah biome. The browse samples were air dried at room temperature for 7 days. All samples were then ground to pass a 0.2 mm sieve and kept in a plastic container pending chemical analyses. For all procedures, the tests (qualitative and quantitative analyses) were repeated three times for accuracy.

### 4.3. Cold Extraction of Secondary Metabolites from Browse Plant Species

Approximately 10 g of finely ground browse leaves powder was weighed into a separate 250 mL conical flasks and then about 100 mL of each solvent (methanol, distilled water) was added to the flask and sealed with laboratory film (parafilm, Bemis, Brampton, ON, Canada). The flasks were kept at normal room temperature for about 7 days, and the flasks were daily shaking for a period of 10–15 min. After 7 days, the extracts were then filtered by using Macherey-Nagel No.1 filter paper under vacuum and then air dried at room temperature in a Petri dish. The weight of each Petri dish was noted prior to drying of the extracts. The weight of the extracts was then calculated from the difference following the procedure demonstrated by Harborne [68].

### 4.4. Preliminary Qualitative Phytochemical Screening of Crude Methanolic and Water Extract

Qualitative phytochemical analyses were carried out using the cold extracts from browse plants following the standard procedure [69] to identify the constituents as demonstrated by Harborne [68], Trease and Evans [70], and Sofowora [71]. The tests for the presence or absence of various plant secondary metabolites (PSM) are summarized below.

#### 4.4.1. Test for Saponins

About 0.5 g of the crude plant extracts was dissolved with distilled water (10 mL) in a test tube. The suspension was shaken in a test tube for about 10–15 min. A 2 cm layer of foam was taken as preliminary evidence for saponins [72]. The layer of foam was used to determine the concentration level of saponins, whereby the higher level of foam indicated a higher level of saponins.

#### 4.4.2. Alkaline Reagent Test (Flavonoids)

Alkaline reagent test: Exactly 0.5 g of plant extract was firstly dissolved with solvent (methanol and distilled water) up to 10 mL in a test tube, and, secondly, an aqueous solution of the extracts was then treated with 10% ammonium hydroxide solution. The yellow fluorescence indicates a positive test for flavonoids [73,74]. The highest level of yellow fluorescence in the test tube symbolized the reflection of high flavonoids.

#### 4.4.3. Test for Tannins

About 0.5 mg of crude plant extracts was added to 10 mL of their freshly prepared solvents (methanol, distilled water) in a test tube and shaken to dissolve. A few (2–4) drops of 0.1% ferric chloride was added and produced brownish green or a blue-black coloration, which confirms the presence of tannins [69,74]. The darkest blue-black color was regarded as a reflection of high concentration of tannins.

#### 4.4.4. Test for Phenols

Ferric chloride test: Aqueous plant extracts were treated with 3–4 drops of 0.1% ferric chloride solution. The formation of a bluish green or black color was confirmed as a positive presence of phenols [75]. The darkest black color was regarded as a reflection of high concentration of phenols.

#### 4.4.5. Test for Cardiac Glycosides

Keller–Killani test: About 0.5 g of crude plant extract was dissolved with 5 mL of distilled water in the test tube and shaken. Each 5 mL extract in the test tube was treated with 2 mL of glacial acetic acid containing a few drops of 0.1% ferric chloride solution. This was underplayed with 1 mL of concentrated sulfuric acid (H_2_SO_4_) along the side of the test tube. The formation of a brown ring at the interface indicates deoxysugar characteristics of cardenolides, which gives a positive presence of cardenolides. The appearance of a violet-green ring below the brown ring in the acetic acid layer indicates a positive result for cardiac glycosides [76].

#### 4.4.6. Test for Terpenoids

Salkowski test: The crude plant extract (5 mL) was separately shaken with 2 mL of chloroform and followed by careful addition of concentrated sulfuric acid (2 mL) along the test tube to form a layer. A reddish brown coloration of the interface formed, confirming a positive presence of terpenoids [69,76,77].

#### 4.4.7. Test for Phlobatannins

About 2 mL of extract of each plant sample was boiled with 1% aqueous hydrochloric acid. Appearance of a red precipitate was taken as evidence for the presence of phlobatannins [69].

### 4.5. Quantitative Phytochemical Analysis

The quantitative assay was carried out using the prepared plant extract as described by Makkar [78]. The extraction preparation is described below, and procedures for determination for phenols, tannins, and condensed tannins are summarized below.

#### 4.5.1. Extract Preparation

Finely ground browse plant material (200 mg) was weighed into a small (25 mL) glass beaker; 10 mL of aqueous acetone (70%) was added, and then the glass beaker was suspended in an ultrasonic water bath (Branson 5800) for about 20 min at room temperature [78]. The content in the glass beaker was then transferred into small centrifuge tubes and subjected to centrifugation for about 10 min at speed of 3000 rpm, with temperature set at 4 °C. After centrifugation, the supernatant was transferred into large test tubes, and the solid particles were left in the small centrifuge tubes. The collected supernatant was kept on ice for further analyses.

#### 4.5.2. Determination of Total Phenolic Content

A 0.01 mL aliquot of tannin-containing extract was pipetted into a test tube. Distilled water (0.49 mL) and 0.25 mL of the Folin–Ciocalteu reagent were added to the test tube and then followed by 1.25 mL of sodium carbonate solution [79]. The test tubes were then vortexed (Lasec Vortex Mixer) and then allowed to stand for 40 min at room temperature. The absorbance was recorded at 725 mm wavelength using a spectrophotometer (T60 UV-Visible Spectrophotometer, PG Instruments). A standard curve was developed using tannic acid (0.5 mg/mL), and concentration of tannic acid equivalent in browse leaf samples was predicted from this curve. Concentration of tannic acid in dry matter was expressed as [Tannic acid in grams/Dry matter%] × 100.

#### 4.5.3. Determination of Total Tannins

Polyvinyl-pyrrolidone (PVP) was used to bind tannins for determining total tannins (TTs), as demonstrated by Makkar [78]. A 100 mg sample of polyvinyl pyrrolidone was weighed into a glass laboratory test tube; 1.0 mL of distilled water and 1.0 mL of the tannin-containing extract were added, and this was stored at a temperature of 4 °C for about 10–15 min. Afterwards, it was centrifuged at 3000 rpm for 10 min. Subsequently, the supernatant was collected with a simple phenolic other than tannins, and the absorbance or UV reading was recorded as mentioned above and expressed as tannin acid equivalent (TAE). Total tannin content was calculated as a difference between simple phenols and total phenol (TP) content in the extract.

#### 4.5.4. Determination of Condensed Tannins

The same acetone extract used in the soluble phenolics assay was used to assay for soluble/extractable-condensed tannins (SCT) using the butanol-HCl reagent (95:5 *v*/*v*) [80]. Aqueous acetone extract (0.5 mL) was pipetted into a glass screw cap test tube and 5 mL butanol-HCl reagent added. The test tube was closed and then placed on a heating block at 100 °C for 1 h. Absorbance was measured after the test tubes had cooled to room temperature. The measurements were reported as absorbance units (au) per 200 mg sample at 550 nm wavelength using the spectrophotometer as prescribed by Makkar [78] (Appendix A).

### 4.6. Statistical Analysis

Data on quantitative phytochemical composition (insoluble phenols, soluble tannin, and soluble condensed tannin) were analyzed using two-way analysis of variance (ANOVA) that used general linear model (GLM) procedures of SAS 9.3 [81] for a factorial treatment arrangement in a completely randomized design (CRD). Factors such as browse species and soil type were considered in this study. The following general linear model was used:Yijk=µ+Bi+Lj+(B×L)ij+εijk
where *Y_ijk_* is the dependent variable, *µ* is the overall mean, *B_i_* is the effect of browse species, *Lj* is the effect of two different locations (soil type), (*B × L*)*_ij_* is the interaction effect between browse species and two different locations, and *ε_ijk_* is the random error term associated with observation *ijk* and assumed to be normally and independently distributed. Where significant variation was detected, multiple comparisons of treatment means were carried out using the probability of difference (pdiff) option of the GLM procedure of SAS [81]. In addition, multivariate analysis was used to test soil type and species effect on soluble phenols, insoluble tannins, and condensed tannins of woody species found in AKS-CH and GM-L soil types.

## 5. Conclusions

The methanol and distilled water extracts of browse species examined in this study contain a variety of phytochemicals that could be responsible for their medicinal, anti-microbial, and antioxidant properties. The most important chemicals are saponins, alkaloids, flavonoids, glycosides, phlobatannins, and terpenoids, which have a metabolic role in biological systems and perform a protective role in livestock. Furthermore, most of the species analyzed had low concentrations of chemical constituents that are essential for maintaining good health. Though there were some discrepancies, the findings suggest that different sites (soil type) and species affected the phytochemical constituent content. To enhance and maximize utilization of these browse plant species, there is a need for various studies to identify all other bioactive compounds constituted in these plant species that may have detrimental effects on animals, e.g., alkaloids, glucosinolates, and terpens. All browse species exposed to animals can be subjected to screening to determine the concentration level of phytochemicals. Multivariate analysis tests revealed a significant effect of both factors and their interaction on soluble phenols, insoluble tannins, and condensed tannins of woody species. To those browse species (*S. caffra*, *V. karroo*, and *D. cinerea*) having high concentration levels of tannins, there is a need to use wood ash or PEG as a remedy to deactivate high tannin protein bonds that reduce the availability of nutrients to the ruminal microbes. The outcomes of this study will provide important information and knowledge on how to improve livestock health and production in semi-arid regions.

## Figures and Tables

**Table 1 molecules-27-01462-t001:** Qualitative phytochemical screening using methanol and distilled water on woody species found in GM-L soil type.

Species	Phytochemical Group													
	Saponins		Flavonoids		Tannins		Phenols		Glycosides		Terpenoids		Phlobatannins	
	W	M	W	M	W	M	W	M	W	M	W	M	W	M
*A. digitata*	++	+	+	-	-	+	-	-	+	+++	+++	+++	-	-
*A. johnsonii*	++	+++	+++	+++	+++	+++	+++	+++	++	+++	++	+++	-	+++
*B. maughamii*	+++	+++	+	-	+	+	-	+	+	+	+	++	-	-
*B. discolour*	+++	+	+++	+++	++	++	-	+	++	++	+++	+++	-	-
*B. zeyheri*	++	+++	+++	++	+++	+++	+++	+++	+	+	+	+++	-	-
*B. mollis H*	+++	+++	++	++	+++	+++	+++	+++	+++	+++	++	+++	-	+++
*Catha edulis*	-	-	+	+	+	-	+	+	+	+	+	+	-	-
*C. mopane*	+++	++	++	++	+	-	+	++	+++	+++	+++	+++	-	+++
*C. Imberbe*	+	+	+++	+++	+++	+++	+++	+++	+	++	+	+++	-	+
*C. molle*	+++	+++	+++	+++	+++	+++	+++	+++	++	+++	++	++	-	++
*C. collinum*	++	++	+++	+++	++	+++	+++	++	++	++	+++	+++	-	++
*D. melanoxylon*	+	+	+	+	+	++	-	++	+	++	+	+++	-	-
*D. cinerea*	+++	+++	+++	+++	++	+++	++	+++	+++	+++	+++	+++	-	+++
*D. mespiliformis*	+++	+++	+	+	++	+++	+++	+++	++	+++	++	+++	-	+
*E. divinorum*	+	+++	+++	++	++	+++	++	+++	+	+++	++	+++	-	+
*G. flava*	+	-	-	-	-	+	-	-	-	++	+	+++	-	-
*G. flavescens*	+++	+	++	++	+	++	+	++	+	++	+++	+++	-	+++
*G. monticola*	++	+	++	++	-	-	-	-	++	+	+	++	-	-
*G. occidentalis*	++	+++	+++	+++	++	++	++	+++	+	+	+	+	-	++
*M. azedarach*	+++	+++	++	+++	-	-	-	-	-	-	++	+	-	-
*P. africanum*	++	+++	+++	+++	+++	+++	++	+++	+	++	++	+++	-	++
*P. maprouneifolia*	++	-	++	+	-	++	-	++	++	++	+	+++	-	+
*S. molle*	+++	+	-	+	-	-	-	-	-	+	+++	+++	-	-
*S. brachypetala*	+++	+++	+	+	++	++	+++	+++	+++	+++	+	++	-	+
*S. birrea*	++	+++	++	+	+++	+++	++	+++	+	+++	+	+++	-	+++
*S. leptodictya*	++	+	-	+	-	-	-	-	+	-	+++	+++	-	-
*S. caffra*	+++	++	+	++	-	++	-	+	+++	++	+++	+++	-	-
*S. galpinii*	+++	-	+	-	-	+	+	+	++	++	-	++	-	-
*S. nigrescens*	+++	+++	++	+	+++	+++	+	++	++	++	++	+++	-	+++
*S. polyacantha*	+++	+++	+	+	-	++	-	+	+	+	+	++	-	+++
*S. madagascariensis*	+++	+	+++	+++	+++	++	+++	++	++	+++	+++	+++	-	-
*T. sericea*	-	++	++	++	+++	+++	+++	+++	+++	+	+++	+++	-	+++
*T. emetic*	+++	-	++	+	++	++	+	+	+	++	++	+++	-	+
*V. hebeclada*	+++	+	-	+	-	-	-	-	-	-	++	+++	-	-
*V. karroo*	+++	++	-	++	-	++	-	+	-	++	+++	+++	-	+
*V. nilotica*	+++	+	+++	+++	+++	++	++	++	+++	+++	+++	+++	-	++
*V. nilotica* subsp. *Kraussiana*	+	+++	+++	+++	+++	+++	+	++	++	+++	+++	+++	-	+
*V. rechmanniana*	+++	+++	++	++	++	+++	++	+++	++	++	+++	+++	-	+++
*V. tortilis*	+++	++	+	++	+	++	++	++	+	+	+	++	-	++
*V. tortilis* subsp. *raddiana*	-	+	+	++	++	++	++	++	++	+++	+++	+++	-	+
*V. infausta*	+++	++	++	++	++	+	++	+	+	+	+++	++	-	-
*Z. mucronata*	++	+	+	+	-	++	++	++	+	+	+	+++	-	+

+++: highly present; ++: moderately present; +: low; -: absent; M: methanol; W: water extract; GM-L: Glenrosa, Mispah, and Lithosols soil types.

**Table 2 molecules-27-01462-t002:** Qualitative phytochemical screening using methanol and water on woody species found in AKS-CH soil type.

Species	Phytochemical Group													
	Saponins		Flavonoids		Tannins		Phenols		Glycosides		Terpenoids		Phlobatannins	
	W	M	W	M	W	M	W	M	W	M	W	M	W	M
*D. cinerea*	+++	+	-	+	+++	+++	+++	+++	+++	+++	+++	+++	-	+++
*D. lycioides*	+++	+++	-	+++	-	-	-	+++	+	+	+++	+++	-	-
*G. flava*	+++	++	-	++	-	+	+	-	+	+++	+	++	-	-
*M. azedarach*	+++	++	++	++	+++	++	+++	+++	+	+	++	+++	-	+
*P. africanum*	+	+++	-	+++	+	++	+	+	+	+	+	+	-	-
*P. velutina*	+++	++	+	++	-	++	-	+	+++	-	+++	++	-	-
*S. molle*	-	-	-	-	+	+	-	+	++	-	+++	+++	-	-
*S. lancea*	+++	+++	+++	+++	+++	+++	+++	+++	++	-	++	+++	-	+++
*S. leptodictya*	+++	++	+	++	+++	++	+++	+++	+++	+++	+++	+++	-	++
*S. pyroides*	++	++	-	++	++	+++	-	++	++	+	++	+++	-	++
*S. caffra*	+++	++	++	++	+++	+++	+	-	+++	+++	+++	+++	-	+
*S. galpinii*	+++	+++	-	+++	-	-	-	-	-	+	+	++	-	-
*S. mellifera*	+++	+++	+++	+++	++	++	++	+++	+	++	+++	+++	-	++
*T. sericea*	+	++	+++	++	+++	+++	+++	+++	++	++	++	+++	-	+++
*V. erioloba*	+++	+++	+++	+++	+++	+++	++	+++	+	+	+++	+++	-	+
*V. hebeclada*	+++	+	+	+	-	-	-	-	-	-	++	+	-	-
*V. karroo*	+++	++	++	++	+++	+++	++	+++	++	++	++	+++	-	+
*V. nilotica* subsp. *Kraussiana*	+	++	+	++	+++	+++	+++	+++	-	++	-	+	-	+++
*V. robusta*	-	++	-	++	-	++	+	++	++	+++	++	+++	-	-
*V. tortilis*	+++	++	-	++	+	++	+	+	-	+	++	+++	-	+
*Z. mucronata*	+++	+++	-	+++	-	++	-	++	+	+	++	++	-	+

+++: highly present; ++: moderately present; +: low; -: absent; M: methanol extract; W: water extract; AKS-CH: Aeolian Kalahari Sand, Clovelly, and Hutton soil type.

**Table 3 molecules-27-01462-t003:** Soil type and species effect on soluble phenols (SPhs—% dry matter), insoluble tannins (ITs—% dry matter), and condensed tannins (CTs—AU_550_/200 mg) of woody species found in AKS-CH and GM-L soil types.

Species	Soluble Phenols		Insoluble Tannins		Condensed Tannins	
	GM-L	AKS-CH	GM-L	AKS-CH	GM-L	AKS-CH
*D. cinerea*	0.1011 ^aA^	0.0969 ^bB^	0.0259 ^efB^	0.0453 ^aA^	0.6664 ^cdB^	2.2258 ^bA^
*G. flava*	0.0801 ^eA^	0.0795 ^dA^	0.0272 ^deA^	0.0220 ^eB^	0.5141 ^eB^	1.4498 ^dA^
*M. azedarach*	0.0207 ^jB^	0.0830 ^cA^	0.0065 ^hB^	0.0341 ^bA^	0.0114 ^iB^	0.4622 ^gA^
*P. africanum*	0.0788 ^eA^	0.0758 ^eB^	0.0339 ^bA^	0.0264 ^dB^	0.8755 ^aA^	0.8746 ^fA^
*S. molle*	0.0377 ^iB^	0.1000 ^aA^	0.0047 ^iB^	0.0184 ^fA^	0.0340 ^iB^	0.2665 ^hA^
*S. leptodictya*	0.0360 ^iB^	0.0564 ^ghA^	0.0037 ^iB^	0.0334 ^bA^	0.0243 ^iB^	1.1434 ^eA^
*S. caffra*	0.0659 ^gA^	0.0514 ^iB^	0.0312 ^cA^	0.0288 ^cB^	0.7131 ^cB^	1.6542 ^cA^
*S. galpinii*	0.0869 ^dA^	0.0385 ^jB^	0.0284 ^dA^	0.0131 ^gB^	0.1294 ^hB^	0.1638 ^jA^
*T. sericea*	0.0908 ^cA^	0.0566 ^ghB^	0.0386 ^aA^	0.0335 ^bB^	0.8088 ^bA^	0.8461 ^fA^
*V. hebeclada*	0.0160 ^kB^	0.0334 ^kA^	0.0041 ^iB^	0.0064 ^hA^	0.0137 ^iA^	0.0138 ^lA^
*V. karroo*	0.0935 ^bA^	0.0582 ^fgB^	0.0206 ^gB^	0.0338 ^bA^	0.2879 ^gB^	2.3270 ^aA^
*V. nilotica* subsp. *Kraussiana*	0.0897 ^cA^	0.0561 ^hB^	0.0386 ^aA^	0.0275 ^cdB^	0.6288 ^dA^	0.5049 ^gB^
*V. tortilis*	0.0598 ^hA^	0.0598 ^fA^	0.0253 ^fA^	0.0230 ^eB^	0.3530 ^fA^	0.0737 ^kB^
*Z. mucronata*	0.0688 ^fB^	0.1009 ^aA^	0.0335 ^bA^	0.0196 ^fB^	0.1590 ^hA^	0.1802 ^iA^
SE	0.00066		0.00056		0.0165	

^a–l^ In a column, means with different lowercase superscripts differ (*p* < 0.05); ^A–B^ in a row, means with different uppercase superscripts differ (*p* < 0.05); GM-L: Glenrosa, Mispah, and Lithosols soil type; SE: standard error.

**Table 4 molecules-27-01462-t004:** Multivariate tests ^a^ on soil type and species effect on soluble phenols, insoluble tannins, and condensed tannins of woody species found in AKS-CH and GM-L soil types.

Effect		Value	F	Error df	*p*-Value
Soil type	Pillai’s Trace	0.992	2160.883 ^b^	3.000	0.001
	Wilks’ Lambda	0.008	2160.883 ^b^	3.000	0.001
	Hotelling’s Trace	120.049	2160.883 ^b^	3.000	0.001
	Roy’s Largest Root	120.049	2160.883 ^b^	3.000	0.001
Species	Pillai’s Trace	2.978	588.200	39.000	0.001
	Wilks’ Lambda	0.000	1091.889	39.000	0.001
	Hotelling’s Trace	1137.717	1536.404	39.000	0.001
	Roy’s Largest Root	696.770	3001.471 ^c^	13.000	0.001
Soil type * species	Pillai’s Trace	2.968	399.117	39.000	0.001
	Wilks’ Lambda	0.000	856.022	39.000	0.001
	Hotelling’s Trace	1214.070	1639.513	39.000	0.001
	Roy’s Largest Root	982.735	4233.321 ^c^	13.000	0.001

^a^ Design, soil type + species + soil type * species; ^b^ exact statistic; ^c^ the statistic is an upper bound on F that yields a lower bound on the significance level; *p*-value: <0.05; GM-L: Glenrosa, Mispah, and Lithosols soil type; Error df: degrees of freedom error.

## Data Availability

All data are available upon request.

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
