# Peer review of "Effect of Soil Type: Qualitative and Quantitative Analysis of Phytochemicals in Some Browse Species Leaves Found in Savannah Biome of South Africa"

_molecules, 2022, doi:10.3390/molecules27051462_

Round 1
Reviewer 1 Report
In general, this manuscript provides extensively qualitative data that represents the relative content of phytochemicals in various plant species. As the authors said this study only provides preliminary information, could the authors add more justification for this study? it was mentioned these plants possessed many health benefits, has it not been reported elsewhere about the content of those selected phytochemicals? why work backward? Could the authors highlight why used qualitative data/analysis and if they are sufficient to extract information for this study? Authors should compare the results within the scope of this study and therefore relative content shall be used.
line 106-107, 110-110, and 112-113: please be consequent with the title and choose to spell out the type of soil used in the title or at the footnote.
suggestion to add multivariate analysis to capture the overview of the effect of soil/harvesting site and different solvents on the content of phytochemicals.
line 377- did the authors use acetone for the extraction instead of water and methanol? and why water and methods were chosen for the extraction? any preliminary data to support? I found that the method for sample preparation was very crude. maybe the authors could provide similar studies that did not remove solvents before analysis.
line 431-432. please remove it as it was not related to this study.
Author Response
TABLE OF COMMENTS AND RESPONSES
We would like to thank the Editor and Reviewers for taking time to go through our research article and providing constructive comments. We have gone through all the comments and our carefully considered responses are appended below. We hope that our responses are satisfactory, however, we stand ready to make further changes, should these be required
|
COMMENT |
RESPONSE |
|
REVIEWER 1 COMMENTS |
|
|
In general, this manuscript provides extensively qualitative data that represents the relative content of phytochemicals in various plant species. |
Thank you for positive appraisal of our work. |
|
As the authors said this study only provides preliminary information, could the authors add more justification for this study? |
Thank you very much the justification has been added. |
|
it was mentioned these plants possessed many health benefits, has it not been reported elsewhere about the content of those selected phytochemicals? why work backward? |
Thank for your concern: Our study was focusing on high number of species (52 browse species found in savannah biome were considered), whereas, Most of the cited articles in this manuscript were either focusing on certain genus species or less number of species were used for our assessments or were run in the countries other than South Africa E.g. Mokoboki et al., 2005 focused on acacia species Rubanza et al., 2007 focused only on three species of which one of the species was part of our study (A. nilotica) Lowry et al., 1996 This was focusing only on plant phenolic |
|
Could the authors highlight why used qualitative data/analysis and if they are sufficient to extract information for this study? |
Thank you for inputs: Phytochemistry is associated with numerous species of secondary metabolites produced in plants by biosynthesis and the natural combination of all these secondary metabolites gives the general beneficial therapeutic effects of that specific plant. This was also highlighted in our last section of introduction Some related study was were conducted by Ismail et al., 2014; Gul et al., 2017, Yadav et al., 2014; Ingle et al., 2017and Senguttuvan et al., 2014. This authors were also cited in this study. |
|
Authors should compare the results within the scope of this study and therefore relative content shall be used. |
Thank you for input: Though we don’t understand the review might need. In order not to repeat the results on discussion section we decided on to focus on the implication of our results based on the methods used. |
|
line 106-107, 110-110, and 112-113: please be consequent with the title and choose to spell out the type of soil used in the title or at the footnote. |
Thanks: Corrected; We have spell out all types of soil found in savannah biome especially where the plants have been harvested from. |
|
suggestion to add multivariate analysis to capture the overview of the effect of soil/harvesting site and different solvents on the content of phytochemicals. |
Thanks for suggestion: Multivariate analyses were added to capture the effect of harvesting site and species on the content of phytochemicals. |
|
line 377- did the authors use acetone for the extraction instead of water and methanol? and why water and methods were chosen for the extraction? any preliminary data to support? I found that the method for sample preparation was very crude. maybe the authors could provide similar studies that did not remove solvents before analysis. |
Thank you for the input: We have used acetone for quantitative analyses extraction instead of water and methanol due to the procedure method we have for quantitative analyses. (Quantificantion of Tannins in tree foliage, laboratory manual). We have used the procedure adopted by Senguttuvan et al., 2014, Ismail et al., 2014 and Yadav et al., 2014. Jamuna Senguttuvan, Subramaniam Paulsamy*, Krishnamoorthy Karthika, 2014. Phytochemical analysis and evaluation of leaf and root parts of the medicinal herb, Hypochaeris radicata L. for in vitro antioxidant activities. Asian Pac J Trop Biomed 2014; 4(Suppl 1): S359-S367. Abeer M. Ismail, Eman A. Mohamed, Marwa R. Marghany, Fatma F. Abdel-Motaal, Ibrahim B. Abdel-Farid, Magdi A. El-Sayed,. 2016. Preliminary phytochemical screening, plant growth inhibition and antimicrobial activity studies of Faidherbia albida legume extracts. Journal of the Saudi Society of Agricultural Sciences (2016) 15, 112–117. Manjulika Yadav, Sanjukta Chatterji, Sharad Kumar Gupta and Geeta Watal. 2014. Preliminary phytochemical screening of six medicinal plants used in traditional medicine. International Journal of Pharmacy and Pharmaceutical Sciences, 6, 5. |
|
line 431-432. please remove it as it was not related to this study. |
Thank you, we have deleted it. |

Reviewer 2 Report
This is a very interesting work and the authors have made an effort to present their results and methodology well.
I suggest that the authors may consider the following points:
- Tables of the qualitative evaluation: As it is, the tables seem to represent a semi-qualitative (or semi-quantitative) evaluation through the classification in “high”, “moderate” and “low” presence of each component. However, there is no evidence as to “how” this classification came to place or if the protocols employed can support it. Kindly provide this information as it would be crucial for the reader to understand how a qualitative evaluation protocol can result into a classification table as described by the authors.
- Sample measurements: It is assumed that all samples were tested in triplicates. If so, please include. If not, also state if the tests have been repeated.
- Regarding the extraction of secondary metabolites: It is well known that many of the secondary metabolites found in plants are sensitive and can be destroyed by exposure to light, heat, air etc. Kindly explain how the extraction protocol was evaluated as appropriate for this study and if the related extraction conditions were taken under consideration as a potential limitation for the overall outcome.
- Please consider elaborating on the protocol employed for the evaluation of CTs (lines 405-410). Is there evidence from this work (or any recent work) that the absorbance units reflect the concentration of the compounds? I strongly suggest that the authors reflect the findings of this protocol with a standard curve (preferably one created in the same experimental conditions as the samples). If there is any available, please consider sharing it in the supplementary data.
- I would kindly invite the authors to revisit and revise their references. Most updated work is essential to be referenced in order to demonstrate the relevance of this work’s findings with current knowledge.
- Line 60: It is stated that “Some of these woody species contain plant secondary metabolites (phytochemicals) which are anti-nutritional elements, which may negatively affect an animal’s health and nutrition” while in the conclusion (lines 433-435) it is stated “Furthermore, because of the presence of various chemicals that are essential for maintaining good health, the browse plant species examined for phytochemical components seemed to have a great potential to act as a source of beneficial medications and as well as to enhance better health status for humans and livestock”. It is my understating that the authors are referring to CTs as potentially harmful in large concentrations, and thus they have been evaluated. As it would be helpful for the reader, kindly consider elaborating on (1) which phytochemicals can have negative effects on animal nutrition and health, (2) if these phytochemicals have been evaluated in the current study (other than CTs) and then (3) combine this knowledge to establish the conclusion. Please also consider that if there is previous knowledge of phytochemicals that can be “dangerous” for animal health and nutrition, should a screening evaluation target them as a first step?
Author Response
TABLE OF COMMENTS AND RESPONSES
We would like to thank the Editor and Reviewers for taking time to go through our research article and providing constructive comments. We have gone through all the comments and our carefully considered responses are appended below. We hope that our responses are satisfactory, however, we stand ready to make further changes, should these be required
|
COMMENT |
RESPONSE |
|
REVIEWER 2 COMMENTS |
|
|
This is a very interesting work and the authors have made an effort to present their results and methodology well. |
Thank you for positive appraisal of our work, we really appreciate it. |
|
Tables of the qualitative evaluation: As it is, the tables seem to represent a semi-qualitative (or semi-quantitative) evaluation through the classification in “high”, “moderate” and “low” presence of each component. However, there is no evidence as to “how” this classification came to place or if the protocols employed can support it. Kindly provide this information as it would be crucial for the reader to understand how a qualitative evaluation protocol can result into a classification table as described by the authors. |
Thanks for the suggestion: Now we have added more information on the categorization of concentration level in the methodology. |
|
Sample measurements: It is assumed that all samples were tested in triplicates. If so, please include. If not, also state if the tests have been repeated. |
Thank you for the question: The tests were repeated three times and we have now included this in the methodology. |
|
Regarding the extraction of secondary metabolites: It is well known that many of the secondary metabolites found in plants are sensitive and can be destroyed by exposure to light, heat, air etc. Kindly explain how the extraction protocol was evaluated as appropriate for this study and if the related extraction conditions were taken under consideration as a potential limitation for the overall outcome. |
Thanks for your inputs: All procedures for the extraction for secondary metabolites were correctly done following the methods prescribed per each compound |
|
Please consider elaborating on the protocol employed for the evaluation of CTs (lines 405-410). Is there evidence from this work (or any recent work) that the absorbance units reflect the concentration of the compounds? I strongly suggest that the authors reflect the findings of this protocol with a standard curve (preferably one created in the same experimental conditions as the samples). If there is any available, please consider sharing it in the supplementary data. |
Thanks for your input: We have now beefed up the tannins methodological section. Also the calibration standard curve will be included as a supplementary materials. |
|
I would kindly invite the authors to revisit and revise their references. Most updated work is essential to be referenced in order to demonstrate the relevance of this work’s findings with current knowledge. |
Thank you for your input: We have now added the recent and updated references in order to link our work with the current knowledge. |
|
Line 60: It is stated that “Some of these woody species contain plant secondary metabolites (phytochemicals) which are anti-nutritional elements, which may negatively affect an animal’s health and nutrition” while in the conclusion (lines 433-435) it is stated “Furthermore, because of the presence of various chemicals that are essential for maintaining good health, the browse plant species examined for phytochemical components seemed to have a great potential to act as a source of beneficial medications and as well as to enhance better health status for humans and livestock”. It is my understating that the authors are referring to CTs as potentially harmful in large concentrations, and thus they have been evaluated. As it would be helpful for the reader, kindly consider elaborating on (1) which phytochemicals can have negative effects on animal nutrition and health, (2) if these phytochemicals have been evaluated in the current study (other than CTs) and then (3) combine this knowledge to establish the conclusion. |
Thanks for the input: We have now modify our conclusion and now reads as follows: The methanol and distilled water extract of browse species examined in this study contain a variety of phytochemicals that could be responsible for their medicinal, antimicrobial, and anti-oxidant properties. The most important chemicals are saponin, alkaloids, flavonoids, glycosides, phlobatannin and terpenoids, which have a metabolic role in biological systems and perform a protective role in livestock. Furthermore, most of the species analysed had low concentration chemical constituents that are essential for maintaining good health. Though there were some discrepancies, the findings suggest that different sites and species affected the phytochemical constituent content. To enhance and maximize utilization of these browse plant species, there is a need for various studies to identify all other bioactive compounds constituted in these plants species that may have detrimental effects to animals e.g. Alkaloids, glucosinolates and terpens. All browse species exposed to animals can be subjected to screening to determine the concentration level of phytochemicals. To those browse species (S. caffra, V. Karroo and D. cinerea) having high concentration levels of tannins, there is a need to use wood ash or PEG as a remedy to deactivate high tannin protein bonds that reduce the availability of nutrients on the ruminal microbes. The outcomes of this study will provide important information and knowledge on how to improve livestock health and production in semi-arid regions. |
|
Please also consider that if there is previous knowledge of phytochemicals that can be “dangerous” for animal health and nutrition, should a screening evaluation target them as a first step? |
Thank you for the question: We have beefed up the conclusion though we were not specific to certain bioactive as we have noticed that most of them or if not all are having the detrimental effect on animals when they’re in high concentration. |
|
This is a very interesting work and the authors have made an effort to present their results and methodology well. |
Thank you for positive appraisal of our work, we really appreciate it. |

Round 2
Reviewer 2 Report
The authors have made a great effort in order to revise the manuscript and the updated version indeed properly reflects the work done.
I suggest that the authors may consider revisiting the text for an extensive check for grammar and syntaxis. It is essential that you convey the message of your study, which often makes sense for the researcher but it’s hard to translate it for the reader to understand. However, please keep in mind that we must at all times produce high-quality scientific texts, which includes proper use of grammar and vocabulary. Following there are some examples of the points you might consider revisiting, but I strongly recommend that the entire manuscript is screened.
- Line 85-87: Please rephrase. There seems to be a problem with syntaxis.
- Line 92-97: Please consider splitting this.
- Line 176-177: Please include reference.
- Line 179-180: Please rephrase. There seems to be a problem with syntaxis.
- Line 187-189: Please rephrase. There seems to be a problem with syntaxis.
- Line 193-198: Please rephrase. There seems to be a problem with syntaxis.
- Line 199: Please revisit the grammar.
- Line 201-202: Is “animal production” proper here? Please revisit.
- Line 209-210: Please revisit the grammar.
- Line 397: H2SO4. Kindly use subscripts for the numbers.
Author Response
TABLE OF COMMENTS AND RESPONSES
We would like to thank the Editor and Reviewers for taking time to go through our research article and providing constructive comments. We have gone through all the comments and our carefully considered responses are appended below. We hope that our responses are satisfactory, however, we stand ready to make further changes, should these be required.
|
COMMENT |
RESPONSE |
|
REVIEWER 2 COMMENTS |
|
|
We have now gone through the whole copy to minimise grammatical and syntaxis errors in order to improve its clarity and readership, however, we stand ready to make further changes, should these be required. |
|
|
Line 85-87: Please rephrase. There seems to be a problem with syntaxis. |
Thanks for the input: We have now rephrased the sentence. |
|
Line 92-97: Please consider splitting this. |
Thank you for the input: We have now created short sentences as shown in track changes. |
|
Line 176-177: Please include reference. |
Thank you, the reference has now been added. |
|
Line 179-180: Please rephrase. There seems to be a problem with syntaxis. |
Thanks for the input: We have now revised the statement. |
|
Line 187-189: Please rephrase. There seems to be a problem with syntaxis. |
Thank you for the input: We have now revised the statement. |
|
Line 193-198: Please rephrase. There seems to be a problem with syntaxis. |
Thank you for the input: We have now revised the statement. |
|
Line 199: Please revisit the grammar. |
Thank you for the input: We have now revised the statement. |
|
Line 201-202: is ‘’animal production’’ proper here? Please revisit. |
Thanks for question: We have now corrected and revised the statement by using animal health in space of animal production. |
|
Line 209-2010: Please revisit the grammar. |
Thank you for the input: We have now revised the statement. |
|
Line 397: H2SO4. Kindly use subscripts for the numbers. |
Thank you for the input: We have now changed the numbers and it now written like this ‘’H2SO4 ‘’ |
